# Investigation on the Potential Application of Na-Attapulgite as an Excipient in Domperidone Sustained-Release Tablets

**DOI:** 10.3390/molecules27238266

**Published:** 2022-11-26

**Authors:** Yuxuan Xiao, Haiyu Zheng, Meng Du, Zhe Zhang

**Affiliations:** 1Key Laboratory of Eco-Environment-Related Polymer Materials, Ministry of Education of China, Lanzhou 730070, China; 2Key Laboratory of Polymer Materials of Gansu Province, College of Chemistry and Chemical Engineering, Northwest Normal University, Lanzhou 730070, China

**Keywords:** attapulgite, sustained release tablets, excipient, hydrogen bonding interaction, dissolution medium

## Abstract

In this study, Na-attapulgite was explored as an excipient to prepare domperidone sustained-release tablets and test them in accordance with United States Pharmacopoeia requirements. Fourier transform infrared spectroscopy (FTIR), X-ray powder diffraction (XRD) and differential scanning calorimetry (DSC) were employed to explore the compatibility between Na-attapulgite and domperidone. The XRD and DSC show no interaction between the drug and Na-attapulgite. The FTIR spectrum indicates a shift in the absorption of N-H in the drug molecule, which can be explained by the hydrogen bonding interaction between the N-H in the DOM molecule and the -OH on the surface of Na-ATP. The diameter, hardness, friability and drug content of the tablets were measured, and they all met the relevant requirements of the United States Pharmacopoeia. In addition, the tablets with Na-attapulgite as excipient exhibit a better release performance within the release time of 12 h. These results demonstrate that the domperidone sustained-release tablets have been successfully prepared by using Na-attapulgite as an excipient. The doping of Na-ATP in domperidone sustained-release tablets improves the cytocompatibility. Moreover, with the increase of Na-ATP content, cells proliferate remarkably and cell activity is significantly enhanced.

## 1. Introduction

Clay minerals are traditionally used as excipients or active substances in solid, liquid and semi-solid pharmaceutical dosage forms for various administration methods, such as diluents, suspending agents, emulsifiers, disintegrants, binders and so on [1,2,3]. Possessing unique physical and chemical properties, such high-performance material is characterized by low price, abundant reserves, high biological safety and high stability. Therefore, it has developed rapidly in drug delivery systems in recent years and has attracted attention from people [4,5,6,7]. Attapulgite (ATP), also known as palygorskite, is a fibrous hydrated magnesium aluminum silicate clay mineral. Because of its chain-layer structure, it possesses unique physical and chemical properties, such as large specific surface area, high chemical stability, strong adsorption, suspension and slow release [8,9]. Due to its excellent performance, ATP is widely used in the construction, food production, textile, pharmaceutical, catalyst, environmental protection and other fields. In particular, ATP features abundant reserves, low price, environmental friendliness and good biocompatibility. For this reason, it is considered to be one of the most promising materials in the pharmaceutical industry [10,11,12,13,14]. In addition, activated attapulgite can be directly used to adsorb enterobacterial endotoxin [15]. Chemically modified attapulgite exhibits unique properties and can be applied as an excellent drug carrier. The CTAB-modified attapulgite addresses the poor outcome of dispersion in solution and the difficulty in compounding with polymers. Therefore, chemically modified attapulgite shows high stability and exhibits excellent release characteristics for drug release [16]. In summary, attapulgite has a massive potential of application in medical services.

However, few studies have focused on the application of attapulgite as an excipient to prepare tablets. Soares et al. [17] evaluated the use of palygorskite as an excipient in anti-tuberculosis drugs. That study revealed that solid dosage forms obtained by the simple physical mixing of materials have higher rates of drug dissolution. In a study conducted by Jin et al. [18], 5-fluorouracil was inserted into the interlayer of attapulgite and montmorillonite, respectively, to prepare a multifunctional compound that can be used to prevent the occurrence of early gastric cancer. In addition, Wang et al. [12] prepared a series of pH-responsive composite hydrogels composed of chitosan-g-poly (acrylic acid)/attapulgite/sodium alginate. As indicated by the release curve, the increase of ATP content in the composite hydrogel can effectively extend the release time of the drug.

As a gastrointestinal motility regulator, domperidone (DOM) can antagonize the dopamine receptors of the central nervous system through the blood-brain barrier, thereby inhibiting the effect of dopamine in the gastrointestinal tract and increasing the pressure of the lower esophageal sphincter. This is conducive to the prevention of nausea and vomiting and to the enhancement of gastrointestinal peristalsis [19,20]. The main dosage forms of domperidone include tablets, microspheres, pellets, microcapsules and emulsions [21,22]. Furthermore, the types of domperidone tablets mainly include orally disintegrating tablets, gastric retention tablets and Sustained-release film-coated tablets. Zayed et al. [23] developed a fast-dissolving oral membrane loaded with domperidone. Initially, the drug is released into the oral cavity and gradually acts on the whole body from the local area, which improves the oral bioavailability and therapeutic effect of the drug. Daihom et al. [24] prepared a complex of domperidone and resinate, which has good fluidity and thermochemical stability for the delivery of gastric retention drugs. The release profile of domperidone shows the characteristics of slow and controlled release. Due to its low water solubility, domperidone has limited oral bioavailability. In addition, it has a short biological half-life of 7 h [25]. The shorter biological half-life is better for developing sustained- and controlled-release formulations.

The production of tablets mainly includes three techniques: wet granulation, dry granulation and direct compression [26]. Each of these techniques has distinct advantages and disadvantages. Direct compression is considered to be the technology of choice for the production of tablets containing heat- and moisture-sensitive drugs. The main advantage of direct compression is that it requires few unit operations and overcomes the instability of active substances that are sensitive to moisture or heat. Compared with wet granulation, it achieves a higher dissolution rate and requires fewer excipients [27].

In view of the above advantages and disadvantages, this study adopts the modification method of sodium carbonate to dissociate attapulgite, and then uses it to load domperidone to improve the bioavailability of the drug. The possible interactions between the drug and Na-ATP were first investigated by DSC, XRD and FTIR spectroscopy. Then, the sustained-release effect of different ratios of Na-ATP as an excipient for domperidone tablets was investigated. Finally, the zero-order, first-order, Higuchi, Korsmeyer-Peppas and Weibull equations were used to fit the release data of various tablets, while the cell viability was measured by the CCK-8 method. It was proved that Na-ATP as excipient has potential value in preparing domperidone sustained release tablets.

## 2. Results and Discussion

### 2.1. SEM and EDS Analysis

Figure 1 shows the microscopic morphology of ATP, Na-ATP and 5% Na-ATP/DOM composites and the elemental mapping of 5% Na-ATP/DOM. The SEM image of ATP (Figure 1a) shows that under the action of van der Waals forces and hydrogen bonds, ATP rod crystals tend to exist in the form of rod crystal bundles and aggregates [28]. For Na-ATP (Figure 1b), many individual rod-like crystals with a length of about 100 nm appeared. Compared with Figure 1a, the modification of ATP by Na_2_CO_3_ has a greater influence on the degree of dispersion of its rod crystal bundles, and Na_2_CO_3_ is a suitable modifier for dispersing the arrangement of ATP rod-like crystals [29]. These results indicate that ATP modified by Na_2_CO_3_ is more suitable for compounding with DOM. As shown in Figure 1c, the Na_2_CO_3_ modified ATP nanorods are covered by many spherical domperidone particles, indicating that many domperidone particles are firmly fixed on the surface of the ATP nanorods with uniform distribution. As shown in Figure 1d–i, the signals of C, N, O, Mg, Al, Si, Cl elements are observed in the element map of 5%Na-ATP/DOM composite with an even distribution of elements, indicating the domperidone particles are evenly distributed on the surface of the Na-ATP nanorods. These results are consistent with the results observed in the SEM image. SEM-EDS analysis results indicated that the composition ratio of C:N:O:Na:Mg:Al:Si:Cl in the 5% Na-ATP/DOM composite was 24.86:2.75:33.94:0.74:4.63:7.46:23.3:2.31, which revealed practical chemical composition in the 5% Na-ATP/DOM composite.

### 2.2. Tablets Properties

In the process of production and transportation, the tablet requires a certain degree of strength and resistance to brittleness to withstand mechanical shock during handling. The USP34 requires that the diameter and drug content determination results of all batches of tablets should be uniform, the friability should be less than 1% and the hardness should be greater than 5 kg. In Appendix A, the diameter of all tablets is in the range of 6.73 to 6.84 mm, and the drug content is in the range of 98.56% to 99.07%, showing good homogeneity. The friability of the tablet ranges from 0.45% to 0.84%, less than 1%, and the hardness ranges from 6.46 to 7.4 kg, more than 5 kg, all of which meet the requirements of the pharmacopoeia.

### 2.3. FT-IR Analysis

Figure 2A shows the FT-IR spectrum of Na-ATP, DOM, 5% Na-ATP/DOM and 10% Na-ATP/DOM. Na-ATP has characteristic peaks at 3614 cm^−1^ and 3545 cm^−1^, which are attributed to the stretching vibrations of Al-OH and Fe-OH, respectively. The absorption band at 3427 cm^−1^ is related to the -OH stretching vibration peak of water adsorbed on the surface of ATP. In addition, the absorption band at 1639 cm^−1^ is assigned to the H-O-H bending vibration peak of the adsorbed water and bound water on the surface of ATP. This Si-O stretching vibration and Si-O-Si bending vibration were observed at 1030 cm^−1^ and 473 cm^−1^. These FT-IR characteristic peaks are consistent with previous findings [30,31]. From the FT-IR spectrum of DOM, the characteristic absorption bands are 3127 cm^−1^ (N-H stretching vibration peak), 1695 cm^−1^ (C=O stretching vibration peak), 1623 cm^−1^ (N-H bending vibration peak), 1487 cm^−1^ (C=C stretching vibration), 1270 cm^−1^ (C-N asymmetric stretching vibration) and 1066 cm^−1^ (C-N symmetric stretching vibration), as observed in [23,32]. As shown in Figure 2B, compared with the FT-IR spectrum of 10%-Na-ATP/DOM, the absorption band of DOM shifts from 3127 cm^−1^ to 3138 cm^−1^, which strongly indicates hydrogen bond interaction between DOM molecules and -OH on the surface of Na-ATP. Figure 2C shows the hydrogen bonding interaction mechanism, which is more favorable for DOM to be loaded on the ATP surface. In addition, the FTIR spectrum of 10% Na-ATP/DOM is the superposition of the FTIR spectrum of Na-ATP and DOM, which indicates that Na-ATP does not alter the drug molecule.

### 2.4. XRD Analysis

The XRD patterns of ATP, Na-ATP, 5% Na-ATP/DOM and 10% Na-ATP/DOM composites are shown in Figure 3. The characteristic peaks at 2θ = 8.48°, 19.83°, 20.86°, 27.94° and 35.12° are attributed to the characteristic reflections of the 110, 040, 121, 400 and 161 crystal planes of attapulgite. In addition, the characteristic peaks at 2θ = 26.64° and 30.88° are assigned to the characteristic reflections of the quartz and dolomite crystal planes [33,34,35]. The Na-ATP sample showed almost the same XRD pattern, indicating that the crystal structure was not affected by the Na_2_CO_3_ treatment. These results are consistent with previous reports of sodium salt modification [36]. More importantly, the modification of Na_2_CO_3_ and the loading of domperidone had no impact on the XRD pattern of attapulgite due to surface drug molecule binding and the absence of intercalation [31].

### 2.5. Thermal Analysis

ATP and Na-ATP show similar thermal behavior when heated, and there are two different endothermic events (Figure 4): an endothermic peak appears at about 80 °C, which is related to the loss of physically adsorbed water on the ATP surface, and another endothermic peak appears at around 196 °C, which can be explained by the loss of bound water and zeolite water in the ATP channel [37,38]. When DOM is heated, an endothermic peak appears at about 260 °C. This is due to the thermal decomposition of DOM, and the chemical bonds of C_2_H_2_, NO_2_, N_2_ and HCl are broken [32]. DSC is further used to carefully check the state of the contained drugs. The DSC curves of 5% Na-ATP/DOM and 10% Na-ATP/DOM showed a strong exothermic peak at 150 °C. This sharp exothermic peak of domperidone indicates that it exists in the form of crystals on the surface of ATP nanorods. The state of drug existence was not affected during the preparation of the composite material, which was confirmed by the results of the XRD analysis [6].

### 2.6. Drug Release Kinetics Study

The results of the in vitro dissolution study are shown in Figure 5. It was observed that the cumulative drug release rates within 1 h were 73% and 76% for the R3 and R4 dosage forms, respectively, while the cumulative drug release rates for the R1 and R2 dosage forms were as low as 29% and 34%. After 8 h of drug release, the drug release rates of R2, R3 and R4 dosage forms reached equilibrium at 81%, 80% and 80%, indicating almost complete drug release. The results of various tablet evaluation trials showed that of the four dosage forms prepared, the R1 and R2 dosage forms were more effective than the R3 and R4 dosage forms. In addition, the R1 and R2 dosage forms produced better sustained-release effects. This can be explained by studying the hydrogen bonding between the -OH on the ATP surface and the N-H in the drug molecule [39]. Due to the interaction of hydrogen bonds, in the release medium of pH = 1.2, a small amount of Na-ATP/DOM is more conducive to controlling the release of the drug. As the content of Na-ATP/DOM in the tablet increases, however, a large amount of Na-ATP is broken down, causing the tablet to disintegrate too fast, and the release rate increases accordingly [40,41].

Considering the coefficient of determination (R^2^) value of the experimental model measured in Appendix A, the release curve of R1, R2, R3 and R4 dosage forms fits best with the Weibull model. This model is suitable for almost all types of dissolution curve fitting, but it is more suitable for comparing the drug release curves of different types of matrices [42,43]. The relationship between the value of b in the Weibull equation and the drug diffusion mechanism is as follows. When b is less than 0.75, the drug release mechanism follows Fick diffusion law; yet, when b is between 0.75 and 1, the release mechanism follows the random diffusion mechanism. Furthermore, the Peppas equation can be used to describe the entire release process, and the larger the value of b, the smaller the disorder of the release medium [44]. Moreover, as shown in Figure 6, the Weibull model has a high R^2^ value, and the release index (b) is consistently less than 0.75, indicating that the drug release of the four types of tablets follows the Fick diffusion mechanism [45,46].

### 2.7. Cell Viability Analysis

Natural ATP nanorods are often used as an important component of tissue engineering scaffolds to improve the biocompatibility and mechanical properties of scaffold materials [47,48], and recent findings have shown that attapulgite doping can make attapulgite Stone/polymer fiber meshes, hydrogels and 3D-printed composites promote cell differentiation and human bone regeneration [48,49,50]. In order to explore the cytocompatibility between the excipient and the original drug, Figure 7 shows the activity test of DOM, Na-ATP and 5% Na-ATP/DOM on GES-1 cells. The concentrations of DOM and Na-ATP were in the range of 20–100 µg/mL, showing good cytocompatibility, and there was no significant change in cell viability with increasing concentrations, indicating that attapulgite is a biologically safe clay that causes no harm to people. The cell viability of the 5% Na-ATP/DOM composite material was significantly improved, indicating that the sustained-release tablet triggered the proliferation of cells in the presence of clay, which is consistent with previous reports [51]. Based on the above description, the prepared domperidone sustained-release tablet has good cytocompatibility.

### 2.8. Analysis of Drug Sustained-Release Principle

The slow-release mechanism of the drug is shown in the graphic abstract, mainly including the following two points:The modification of Na_2_CO_3_ makes ion exchange occur on the surface of ATP, which is negatively charged as a whole, and the rod crystals repel each other, making them dissociated to a certain extent, which increases the contact probability between the rod crystals and drug molecules.Through ball milling, there is hydrogen bond interaction between the DOM molecule and -OH on the Na-ATP surface. This hydrogen bonding interaction will become stronger with the increase of Na ATP content, which is more conducive to the slow release of drugs.

## 3. Materials and Methods

### 3.1. Materials

Attapulgite clay purchased from Xuyi OuBaite clay material Co., Ltd. (Hauian, China) NaCl (AR) was provided by the Guangdong Chemical Reagent Engineering Technology Research and Development Center (Guangdong, China). Na_2_CO_3_ (AR) was provided by Shanghai Saen Chemical Technology Co., Ltd. (Shanghai, China). Magnesium stearate, pre-cured starch and microcrystalline cellulose (MCC) were all provided by Qufu Tianli Pharmaceutical Excipients Co., Ltd. (Qufu, China). Domperidone (DOM) was provided by Beijing Pharmaceutical Research and Development Center (Beijing, China). All other materials are analytical grade and can be used directly. The PH value of the simulated artificial gastric juice is measured with a PB-10 pH meter (Sedori Corporation, Deggendorf, Germany). AHD2004W electric mixer was used to stir, a TDP-1 pure electric turbo single-punch tablet press (Shanghai, China) was used to press tablets and a UV-2800A UV-visible spectrophotometer (Unico Instruments, Shanghai, China) was used to measure the concentration of the liquid medicine.

### 3.2. Methods

#### 3.2.1. Modification of ATP

First, 10 g of ATP and 1 g of Na_2_CO_3_ were dissolved in 190 mL of water and stirred by using an electric stirrer for 6 h. ATP accounts for 5% of the total solution mass, and Na_2_CO_3_ accounts for 5% of the ATP mass. Then, the sand and stone deposited in the beaker were filtered and washed with deionized water to neutrality. Subsequently, the mixture was dried in a vacuum drying oven at 70 °C for 24 h. Finally, it was ground and filtered with a 200-mesh sieve to obtain Na_2_CO_3_-modified ATP, which was labeled Na-ATP.

#### 3.2.2. Preparation of Powder and Direct Compression

The Na-ATP and DOM passing through a 200-mesh sieve were ball milled at a certain ratio for 2 h to obtain a composite material of Na-ATP and DOM. The ratio of materials is shown in Appendix A. Then, the microcrystalline cellulose, polyvinyl pyrrolidone, magnesium stearate and pre-crossed starch were added at a certain ratio for grinding and dried in a constant-temperature drying oven at 60 °C for 4 h. Finally, a single punch tablet machine was employed to make tablets, which were dried at 60 °C for 12 h. The preparation process is shown in Figure 8. In the above process, the complex of Na-ATP and DOM was labeled as 5% Na-ATP/DOM, 7.5% Na-ATP/DOM, 10% Na-ATP/DOM and 12.5% Na-ATP/DOM. In addition, the dosage forms of the corresponding tablets were denoted as R1, R2, R3 and R4, respectively.

#### 3.2.3. SEM and EDS Measurements

The surface morphology of ATP, Na-ATP and 5% Na-ATP/DOM were measured with a SU8010 (Hitachi, Tokyo, Japan) scanning electron microscope, and the element distribution of the surface micro-zone of Na-ATP was measured with an energy dispersive spectrum elemental composition analyzer.

#### 3.2.4. Friability Measurement

The friability test was carried out in accordance with the regulations of the USP34 Pharmacopoeia [52]. A total of 10 tablets larger than 0.65 g were selected, accurately weighed, transferred to the friability tester (Erweka TAR20, Heusenstamm, Germany) and rotated 100 times. After being taken out, the powder obtained from the tablet was blown off with a hair dryer and weighed accurately. The formula for calculating the percentage of fragility is as follows.
Percentage friability = [(W − W_2_)/W_1_] × 100%
where W_1_ and W_2_ are the weights of the tablets before and after the friability test, respectively.

#### 3.2.5. Hardness and Diameter Measurement

Tablet hardness is the force that compresses along the diameter to break the tablet. The tablet hardness tester (Erweka TBH30, Heusenstamm, Germany) was set to a tablet diameter of 7 mm, in continuous mode; 6 randomly sampled domperidone sustained-release tablets were placed in the hardness tester in turn; and the average tablet hardness was recorded. The diameter of the tablet was measured by a calibrated Vernier caliper.

#### 3.2.6. Drug Content

Ten tablets were crushed, the drug content was determined, and the average value was taken. The tablet powder equivalent to 100 mg of domperidone was mixed with 100 mL of artificial gastric juice (pH = 1.2) and shaken for 30 min. Then, the mixture was filtered through a 0.45 µm micro-porous filter membrane, diluted 50 times and analyzed using an ultraviolet-visible spectrophotometer (UV-2800A, Unico Instruments, Shanghai, China) at 284 nm.

#### 3.2.7. Fourier Transform Infrared (FT-IR) Spectroscopy

FT-IR spectra of ATP, Na-ATP, DOM, 5% Na-ATP/DOM and 10% Na-ATP/DOM were tested by Fourier transform infrared spectrometer (FTS-3000, DIGILAB). Fourier infrared spectroscopy was used to analyze the microscopic groups of the test sample. Through KBr for tablet compression, the FT-IR spectrum was recorded in the range of 4000~400 cm^−1^ with a resolution of 4 cm^−1^.

#### 3.2.8. X-ray Powder Diffraction Method

The XRD patterns of ATP, Na-ATP, DOM, 5% Na-ATP/DOM and 10% Na-ATP/DOM were tested on the Smartlab X-ray powder diffractometer (Rigaku, Japan). The test conditions are as follows: the scanning angle 2θ was within the range of 5~80° and Cu Kα rays were used in continuous scanning mode; scanning was performed at a speed of 5° min^−1^; and the operation was conducted at 40 kV and 40 mA. All samples were measured on a fixed sample stage at room temperature.

#### 3.2.9. Differential Scanning Calorimeter (DSC) Measurements

DSC analysis of ATP, Na-ATP, DOM, 5% Na-ATP/DOM and 10% Na-ATP/DOM was carried out using a differential scanning calorimeter (TA Instruments, Newcastle, DE, USA). Then 6 mg of sample was placed in an aluminum pan and heated at a rate of 10 °C/min in the range of 30 to 300 °C. The temperature was maintained at a constant temperature for 1 min, and then it was reduced to 30 °C at a rate of 10 °C/min. The N_2_ flow rate was 50 mL/min. The obtained thermal data were analyzed by STARe Software v12.10.

#### 3.2.10. Drug Release Experiments

In the USP Dissolution Apparatus II (NTR-6100A, Toyama Sangyo Co., Ltd., Osaka, Japan), the in vitro dissolution of the tablet was studied by using a paddle agitator at 50 rpm and 25 mm depth, with 900 mL of artificial gastric juice (pH = 1.2) as the dissolution medium at a 37 ± 0.5 °C condition. Next, 5 mL of supernatant was taken at intervals of 1 h, 2 h, 4 h, 6 h, 8 h, 10 h and 12 h, and then replaced with the same amount of artificial simulated gastric juice. The aliquot of the supernatant was filtered through a 0.45 µm micro-porous filter membrane, and the drug content was analyzed using an ultraviolet-visible spectrophotometer (UV-2800A, Unico Instruments, Shanghai, China) at the maximum absorption wavelength of 284 nm. The cumulative percentage of the released drug was calculated from the drug concentration. 

#### 3.2.11. Mathematical Modelling of Release Profiles

The drug release data were fitted by zero-order, one-media, fitting, Higuchi, Korsmeyer-Peppas and Weibull equations (Appendix A). The correlation coefficient R^2^ reflects the degree of fit of the release model [53,54].

#### 3.2.12. Cell Viability Assay

The cytotoxic activity of the DOM, Na-ATP and 5% Na-ATP/DOM formulations on normal human gastric epithelial cells (GES-1) was determined by the CCK-8 method. The cells were placed in complete nutrient medium RPMI-1640, 37 °C, 5% CO_2_, humidified and seeded in a 96-well plate at a seeding density of 5000 cells/well. After 12 h, the cells grew 80% to 90%. In a fresh medium, three samples were diluted with different concentrations (20, 40, 60, 80 and 100 μg/mL) and added to the wells, and three replicate wells were set for each sample concentration. The cells in the control group only grew in the medium without drug addition. After culturing for 24 h, they were washed twice with Hank’s solution, and 10 μL of CCK-8 solution was added to each well and mixed well. The control group cells were then incubated in a cell incubator at 37 °C for 1.5 h before being taken out for absorbance measurements. These experiments were performed in triplicate. After the experiment was complete, the percentage of cell viability was calculated according to the procedure described earlier.

## 4. Conclusions

In conclusion, the domperidone sustained-release tablets are successfully prepared with Na-ATP as excipient, which have been characterized by SEM, FTIR, XRD and DSC. The SEM results show that domperidone molecules were evenly distributed on the attapulgite. According to FTIR, XRD and DSC results, Na-ATP did not interact with drugs. In addition, the tablets containing different percentages of Na-ATP, showing immediate and slow release of the drug. Therefore, the dissolution behavior of the drug is related to the content of Na-ATP in the tablet and the pH of the dissolution medium. The results of the cell viability assay show that Na-ATP had good cytocompatibility and could improve cell viability. Therefore, as an excipient, Na-ATP has potential value in preparing domperidone sustained-release tablets.

## Figures and Tables

**Figure 1 molecules-27-08266-f001:**
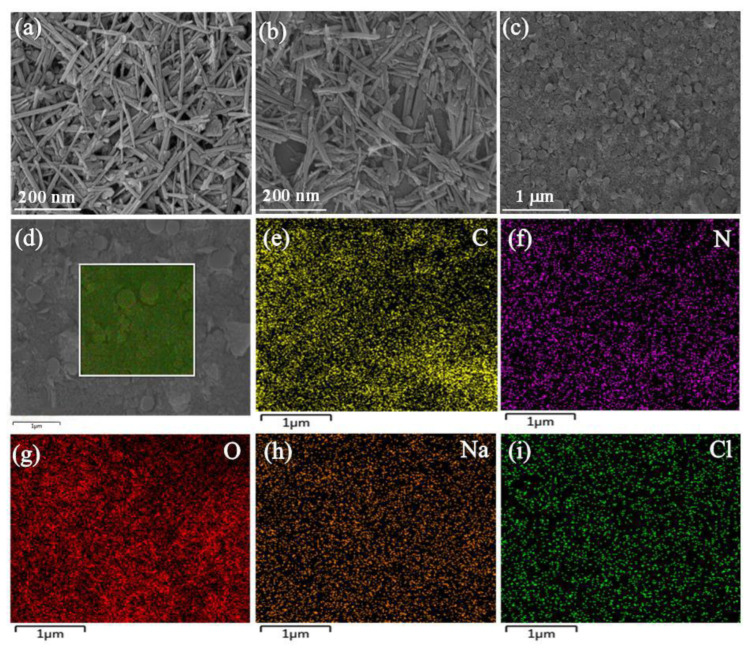
SEM images of (**a**) ATP, (**b**) Na-ATP and (**c**) Na-ATP/DOM; and (**d**) SEM image of selected area of 5%Na-ATP/DOM; and the elemental mapping of (**e**) C, (**f**) N, (**g**) O, (**h**) Na, (**i**) Cl elements.

**Figure 2 molecules-27-08266-f002:**
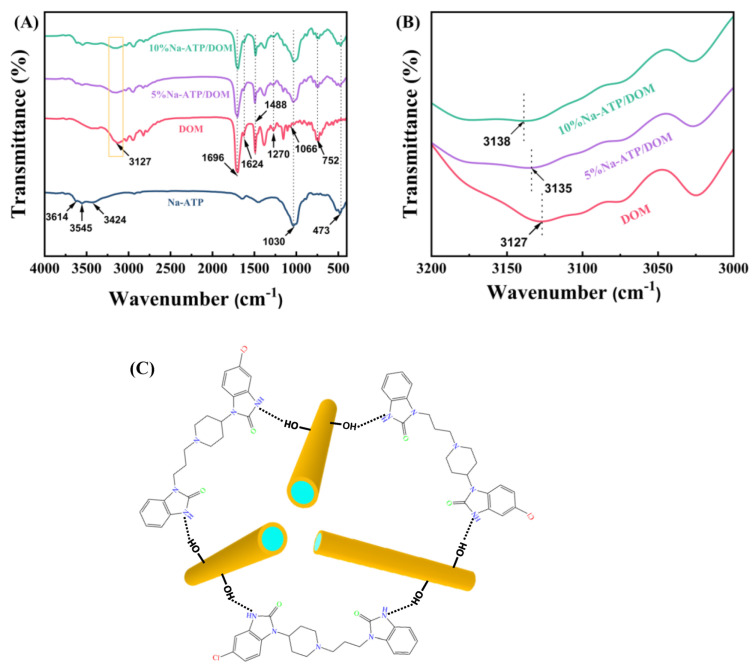
(**A**) FTIR spectrum of Na-ATP, DOM, 5%Na-ATP/DOM and 10%Na-ATP/DOM; (**B**) amplified spectrum of the marked area in (**A**); (**C**) Mechanism of hydrogen bond interaction between DOM molecules and Na-ATP.

**Figure 3 molecules-27-08266-f003:**
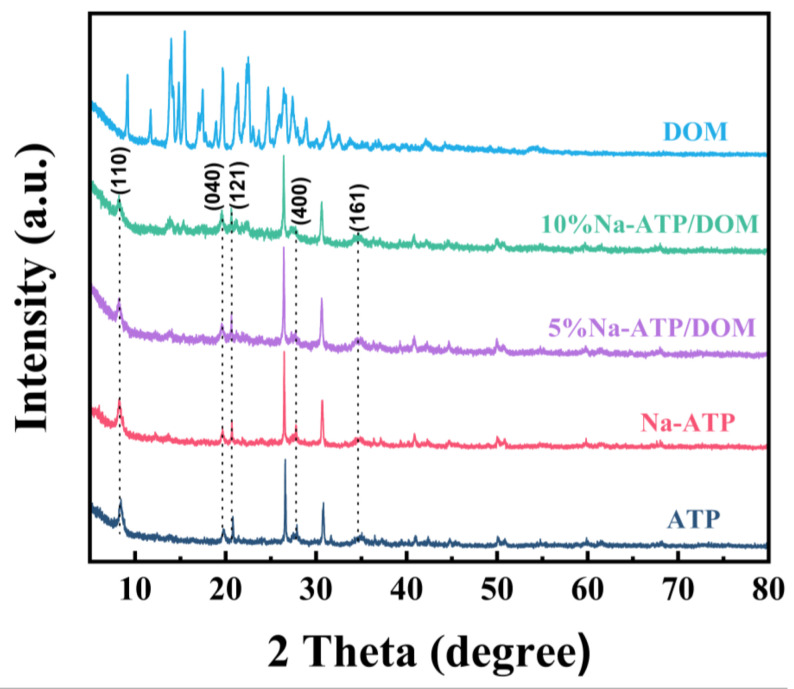
XRD curves of ATP, Na-ATP, DOM, 5% Na-ATP/DOM and 10% Na-ATP/DOM.

**Figure 4 molecules-27-08266-f004:**
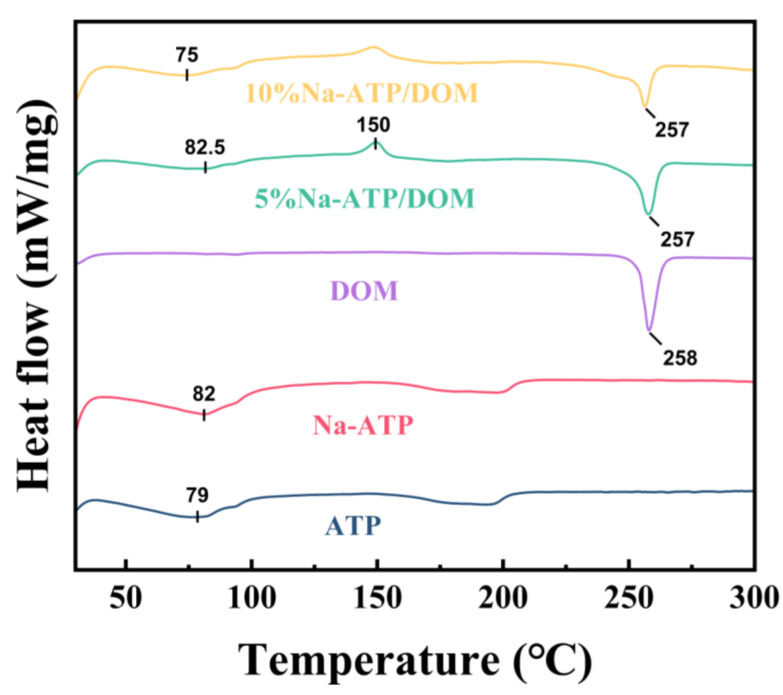
DSC curves of ATP, Na-ATP, DOM, 5% Na-ATP/DOM and 10% Na-ATP/DOM.

**Figure 5 molecules-27-08266-f005:**
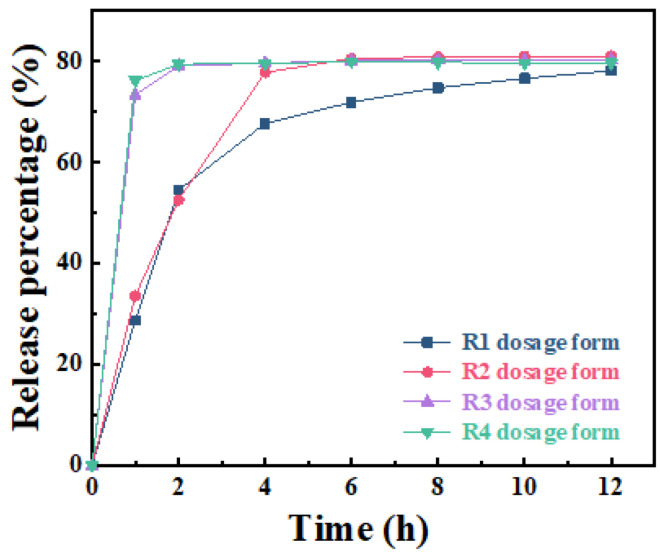
Cumulative drug release rate for R1, R2, R3 and R4 dosage forms.

**Figure 6 molecules-27-08266-f006:**
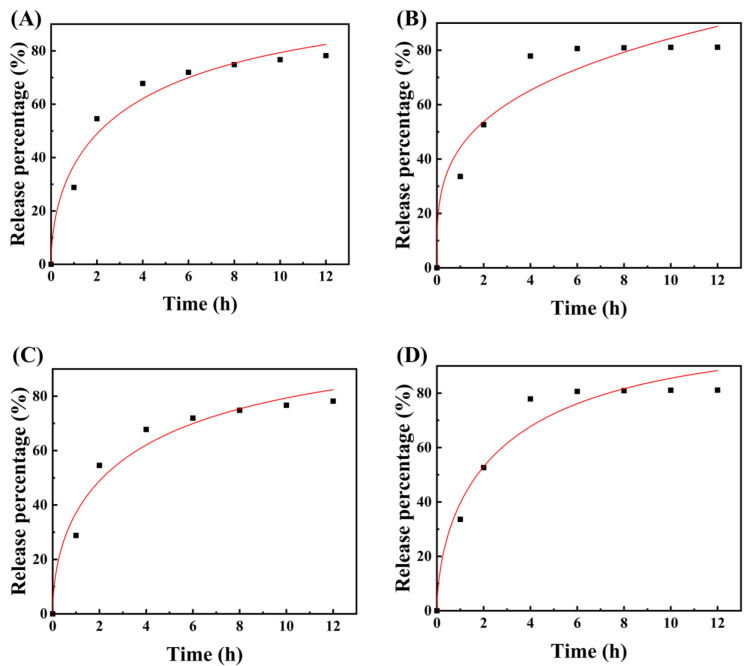
The drug release curves of R1 and R3 dosage forms were fitted by Korsmeyer-Peppas (**A**,**B**) and Weibull (**C**,**D**) models, respectively.

**Figure 7 molecules-27-08266-f007:**
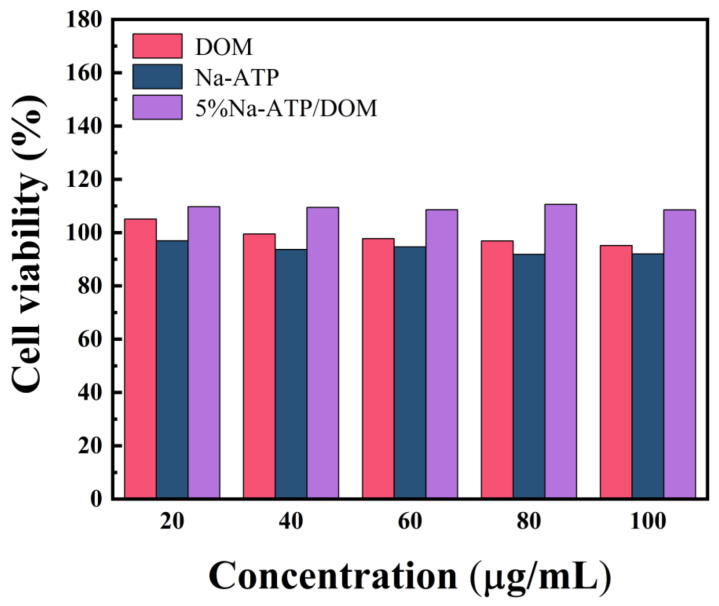
Cell viability of DOM, Na-ATP and 5% Na-ATP/DOM dosage forms at concentrations ranging from 20 ug/mL to 100 ug/mL after 24 h.

**Figure 8 molecules-27-08266-f008:**
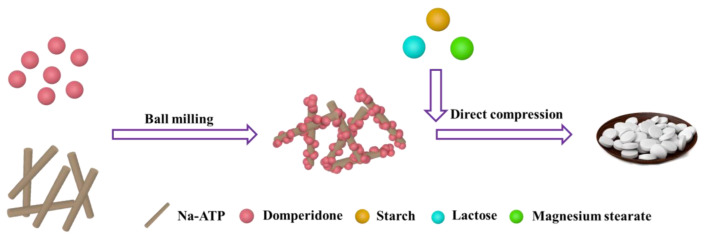
Preparation of domperidone sustained-release tablets.

## Data Availability

Not applicable.

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
