# Peer review of "Investigation on the Potential Application of Na-Attapulgite as an Excipient in Domperidone Sustained-Release Tablets"

_molecules, 2022, doi:10.3390/molecules27238266_

Round 1

Reviewer 1 Report

The manuscript “Investigation on the potential application of Na-attapulgite as “an excipient in domperidone sustained-release tablets” adopts the modification method of sodium carbonate to dissociate attapulgite, and then use it to load domperidone to improve the bioavailability of the drug. In this study, Na-attapulgite was explored as an excipient to prepare domperidone sustained release tablets and test them in accordance with US Pharmacopoeia requirements. 

The paper contains interesting results, and it appears to me that the authors have done a systematic testing procedure. However, there are major issues that need to be addressed before this work can be considered for publication. 

1-     The language of the manuscript is generally good. However, some grammatical errors are observed in different sections of the manuscript. These need to be addressed. 

2-     This manuscript sounds more like a report and not a scientific article. In this manuscript, the results are discussed but the analysis and discussion are so weak. More emphasis is needed to be applied to the discussion and analysis of the obtained results. 

3-     The authors claim that the domperidone sustained-release tablets prepared in this work are in accordance with the US Pharmacopoeia requirements. This is, in my mind, a big claim. If this is the case, it should be clearly discussed (in the introduction) what those requirements are and what testing procedures should be done or followed to evaluate the developed product. The results need to be compared with the criteria and then the conclusions should be made.  

Reviewer 2 Report

Congratulations on the good work, please find my minor suggestions/remarks here:

Content:

Is the coefficient of determination a good choice to differentiate between non-linear models? It is instead suggested to use AIC or BIC for this. Could Table S4 report the RMSE values and one of the criteria (AIC or BIC) instead or additionally to the R² please?

Why does the dissolution profile reach maximum levels of only 80% please? 

Spelling/grammar:

Line 75: This shorter biological half-life will be more efficiently administered in sustained and controlled release formulations.

Paragraph at line 77: The last two sentences of the paragraph are repeated.

Line 95: … which proved that Na- 95 ATP can be used as an excipient to prepare domperidone sustained-release tablets. Potential

Line 254: how is this claim of superposition substantiated?

Line 306: coefficient of determination (R2 R²)

Line 331: Safe does not need to be capitalized.

Reviewer 3 Report

Reviewing of the article entitled “Investigation on the potential application of Na-attapulgite as an excipient in domperidone sustained-release tablets”.

Summary

The present article relates the fabrication and characterization of sustained release tablets made of Na-attapulgite (Na-ATP) as the excipient and domperidone (DOM) as the active pharmaceutical ingredient. Both the structural characterization of the Na-ATP/DOM tablets and the efficiency of the drug product are thoroughly investigated in the present article. It appears that DOM preferentially adsorbs on the surface on the rod-like crystals of Na-ATP with no noticeable intercalation into the Na-ATP molecular sheets as shown by X-ray diffraction measurements for example. As far as the formulated drug properties are concerned, it is shown that tablets with lower Na-ATP content exhibit the best drug release profiles compared to those with higher Na-ATP content. Finally, the cytotoxic activity of the formulated drug has been assessed on normal human epithelial cells and it appeared that the cell viability is even improved for the Na-ATP/DOM formulated drug. All those results point to a safe and efficient use of Na-ATP as excipient for drug tablets.

My opinion

The research dealt with in the manuscript is well conducted and the conclusions made in the manuscript are well supported by the experimental results. The description of the state of the art is consistently carried out with adequate references. A broad range of characterization techniques is used which enable to fully assess the potentiality of Na-ATP as excipients for domperidone tablets. In particular, the structure and microstructure of the Na-ATP/DOM formulations is examined and the drug properties are also assessed with dedicated experiments. Hence, I recommend this article to be published with only minor modifications (see below).

General questions and comments.

Some subsections of the Methods section are written as in a protocol book with verbs at the active form (example : disperse 10 g of ATP… ; Then filter the sand and stone…). The authors are asked to preferentially use the passive form to describe the various protocols (example : 10 g of ATP have been dispersed …).

Page-by-page corrections / questions

-Introduction, page 3 lines 80-87: the 2 consecutive sentences “The main advantage of using direct compression is that it requires …” and “Compared with wet granulation, …” has been repeated twice. One has to be removed.

-Results and discussion, page 8 figure 4: The caption of figure 4 is missing.

-Results and discussion, page 8 line 282: The authors use “DSC spectra” but a DSC result is not a spectrum. DSC curves or thermograms should rather be used.

-Results and discussion, page 8 line 306: The coefficient of determination is noted R2 while it should have been noted R² (Confusion can be made with the R2 sample).

-Results and discussion, page 8 line 331: “safe” is written with a capital letter “S” while it should not (it is in the middle of a sentence).

-Results and discussion, page 8 line 332: A word is likely missing in the sentence “The cell viability of the 5% Na-ATP/DOM composite material was significantly improved, indicating that the sustained release …” since its meaning is not clear. The authors are asked to correct this sentence.

Round 2

Reviewer 1 Report

the revision performed by the authors answers all my latest comments, I, Thus, accept the manuscript.